# A Novel High-Throughput and Sensitive Electrochemiluminescence Immunoassay System

**DOI:** 10.3390/bioengineering11090885

**Published:** 2024-08-31

**Authors:** Xiancheng Liu, Shuang Zhao, Yan Xu, Bo Zhang, Junli Huang, Feng Liu, Ning Yang, Wenhua Lu, Dandan Shi, Dezhong Xie, Yuanfang Hou, Guixue Wang

**Affiliations:** 1Key Laboratory for Biorheological Science and Technology of Ministry of Education, National and Local Joint Engineering Laboratory for Vascular Implants, Bioengineering College of Chongqing University, Chongqing 400044, China; 20191901696g@cqu.edu.cn (X.L.); zhaos_huang@163.com (S.Z.); aaron.xu@lifotronic.com (Y.X.); huangjunli@cqu.edu.cn (J.H.); 2Key Technology Engineering Laboratory of Immunoassay and Liquid Chromatography In Vitro Diagnosis in Shenzhen, Lifotronic Technology Co., Ltd., Shenzheng 518005, China; bo.zhang@lifotronic.com (B.Z.); feng.liu@lifotronic.com (F.L.); ning.yang@lifotronic.com (N.Y.); wenhua.lu@lifotronic.com (W.L.); dandan.shi@lifotronic.com (D.S.); dezhong.xie@lifotronic.com (D.X.); 3Chongqing Engineering Research Center of Pharmaceutical Sciences, Pharmaceutical College, Chongqing 401331, China; 4JinFeng Laboratory, Chongqing 401329, China

**Keywords:** electrochemiluminescence immunoassay system, system timing module, magnetic separation, compensation system

## Abstract

(1) Background: In vitro diagnostic (IVD) tests are the main means of obtaining diagnostic information for clinical purposes. The electrochemiluminescence immunoassay (ECLIA) has become an important in vitro diagnostic technique. It has unique advantages and broad market prospects due to its sensitivity, detection limit, detection range and reagent stability. At present, there is a need to develop and optimize electrochemiluminescence immunoassay subsystems to achieve high-throughput outputs and accurate detection of instruments to promote their clinical application. (2) Methods: On the basis of the demand for clinical testing instruments with high detection accuracy and speed, this study constructed an electrochemiluminescence immunoassay system by designing magnetic separation modules, introducing PMT and optimizing the system timing regulation capability. (3) Results: The magnetic separation modules can increase the detection accuracy, the PMT system increases the detection sensitivity and optimized system timing can achieve maximum test output. Furthermore, this system performs well in terms of its linearity, detection limit, signal-to-noise ratio, precision and accuracy. (4) Conclusions: The electrochemiluminescence immunoassay system is capable of high throughput with good sensitivity and accuracy, meeting the basic requirements of clinical applications for detection capacity and output throughput.

## 1. Introduction

In vitro diagnostics (IVDs), i.e., in vitro testing of human bodily fluids, tissues and blood to obtain clinical diagnostic information, constitute 80% of clinical diagnoses currently relying on IVDs. There are numerous subdivisions of IVDs, including biochemical diagnostics, immunological diagnostics, molecular diagnostics, microbiological diagnostics, etc., among which immunological diagnostics account for the largest proportion [1]. Chemiluminescence immunoassays are the mainstream method of immune diagnosis and have broad development prospects [2]. With the research on chemiluminescence, the technical routes of chemiluminescence that have been put into clinical application include direct chemiluminescence, enzymatic chemiluminescence and electrochemiluminescence [3,4]. The sensitivity of direct chemiluminescence luminometers is slightly lacking, while enzymatic chemiluminescence has high requirements for enzyme transport and preservation. However, electrochemiluminescence has advantages in terms of sensitivity, detection limits, detection range and reagent stability and has unique advantages and broad market prospects [3,5,6].

The electrochemiluminescence immunoassay (ECLIA) is an advanced technology for detecting ultratrace amounts of active substances. This method combines high-sensitivity chemiluminescence determination, highly controllable electrochemical analysis and highly specific immune reactions [7,8]. As important pieces of chemiluminescence technology, electrochemiluminescence immunoassay instruments have been developed for clinical analysis by Roche, Abbott, Beckman Coulter, Siemens and other companies. High-throughput, automated and intelligent assembly lines represent future directions for the development of chemiluminescence immunoassay instruments. On the basis of the demand for clinical testing technologies with high testing speed and accuracy, electrochemiluminescence immunoassay instruments need to be optimized and innovated to increase the test throughput and improve the sensitivity and accuracy of testing systems.

In this work, to improve the detection accuracy of the assay, a magnetic separation module, which consists of magnetic beads with a uniform-size core–shell structure and a large surface area of magnetic particles and high adsorption efficiency, was introduced into an electrochemiluminescence immunoassay system. A compensation system for current output photomultipliers is important for significantly improving the sensitivity of detection via current supplementation of low signals during the detection process. The Hamamatsu H10721-01 was finally selected for the evaluation of its wavelength sensitivity, dynamic range, dark current, detected light intensity range, etc. Furthermore, a system timing module was personalized for the electrochemiluminescence immunoassay system by using a time-slice rotation scheduling algorithm, customizing the timing according to the principles of the electrochemiluminescence immunoassay and performing timing optimization to achieve the highest throughput. Finally, this study analyzed the overall performance of an electrochemiluminescence immunoassay system with an optimized design, including the linearity, detection limit, signal-to-noise ratio, precision and accuracy, all of which showed good performance and potential for clinical detection. In conclusion, this study achieved high-throughput, sensitive and accurate detection in the electrochemiluminescence immunoassay by optimizing the performance of each system, which will promote the clinical application of electrochemiluminescence immunoassay technology.

## 2. Materials and Methods

### 2.1. Materials

Alpha-Feto Protein (AFP) was purchased from WHO NIBSC (SRM@2921, 1st 72/225). Dynabeads M-280 Streptavidin were purchased from invitrogen (35131D). hs-cTnT-STAT (Hypersensitive cardiac troponin T), adrenocorticotropic hormone (ACTH), fibrin degradation products (FDPs) and other specially labeled reagents and consumables were obtained from Lifotronic Technology Co., Ltd., (Shenzheng, China).

### 2.2. Linear Analysis

Standard samples with high values close to the upper limit of the linear range were diluted to 12 concentrations, and the samples were randomly arranged for determination and calculated according to the formula below.
(1)$$r=\frac{\sum X_{i}Y_{i}-\left(\sum X_{i}\sum Y_{i}\right)/n}{\sqrt{\left[\sum X_{i}^{2}-{\left(\sum X_{i}\right)}^{2}/n\right]\left[\sum Y_{i}^{2}-{\left(\sum Y_{i}\right)}^{2}/n\right]}}$$
where

*r* is the linear correlation coefficient;

*X_i_* is the indicated value of the measured sample or the theoretical value of the diluted sample;

*Y_i_* is the average of the actual measurements of samples from 3 replicate determinations;

*n* is the number of samples measured.

### 2.3. Detection Limit Analysis

When testing the detection limit of the electrochemiluminescence immunoassay system to detect AFP, we prepared two standard products with approximate detection limits and tested them for three days. Four repeated tests were carried out each time, the statistical detection results were analyzed and whether the percentage of ≥LoB declaration (0.2 IU/mL) in the detected results accounted for ≥87% of the total number of detected results was determined. If it did, the detection limit requirements were met. When the detection limit of hs-cTnT-STAT was tested, 5 low-value samples with an approximate detection limit of concentration (LoD ≤ 5 pg/mL) were tested, each sample was tested 5 times, and the test results were sorted by size. The number of test results below the blank limit (LoB ≤ 3 pg/mL) should be less than or equal to 3. The detection limit requirements were met.

### 2.4. Signal-to-Noise Ratio Analysis

The signal-to-noise ratio of the electrochemiluminescence immunoassay system was evaluated via comparison with that of the Roche system. The signal-to-noise ratios were compared for the detection of high- or low-concentration standard substances containing AFP or hs-cTnT-STAT.

### 2.5. Precision Analysis

The standard samples were tested 40 times consecutively, the clinical samples were measured continuously for 20 days and the coefficient of variation (*CV*) was calculated.
(2)$$\bar{X}=\sum X_{i}/n$$
(3)$$\mathrm{Standard}\;\mathrm{deviation}\;S=\sqrt{\sum \left(\bar{X}-Xi{)}^{2}/(n-1\right)}$$
(4)$$\mathrm{Coefficient}\;\mathrm{of}\;\mathrm{variation}\;CV=S/\bar{X}$$

### 2.6. Accuracy Analysis

The accuracy of the electrochemiluminescence immunoassay system was analyzed by comparison with the test results of Roche, a leading international company. First, the accuracy of the detection was analyzed by fitting the correlation between the detection results of the electrochemiluminescence immunoassay system and the results of the Roche instrument. Moreover, the relative deviation, absolute deviation, relative deviation ranking and absolute deviation ranking of the electrochemiluminescence immunoassay system were analyzed to determine the accuracy of the detection results. In this study, the detection accuracies of AFP and hs-cTnT-STAT were analyzed.

## 3. Results

### 3.1. Magnetic Separation Module to Improve Detection Accuracy

Tested samples are often complex and contain the target substance and various other substances, which can influence the test process and reduce the accuracy of the test. To improve the possibility of clinical application of the electrochemiluminescence immunoassay system, it is important to further improve the accuracy of the assay to meet the clinical testing requirements. We therefore designed a magnetic separation module for the system based on permanent magnets and magnetic beads to separate interferents in the sample from the sample–reagent conjugate during analysis to achieve purification and enrichment of the target and eventually improve the precision of sample detection [Figure 1A].

Magnetic beads are a major component of magnetic separation modules, and their design is important for achieving good magnetic separation with magnetic enrichment capabilities. The large surface area of the magnetic particles and the grooves on their surface increase the surface molecular loading (10 μg of biotinylated antibody per mg of streptavidin-modified magnetic bead surface). The high adsorption efficiency of synthetic magnetic particles therefore reduces the use of samples and increases the sensitivity of the assay, providing a basis for clinical applications.

The magnetic separation module is designed to separate the target material from the interfering material and to remove the remaining free fluorescent markers from the reaction system, further improving the accuracy of the analysis system. The introduction of a magnetic separation module can significantly improve the detection capability of the electrochemiluminescence immunoassay system. As shown in Figure 1B, the standard curve of the FDP revealed that when the magnetic separation module was added, the detection signal value of the electrochemiluminescence immunoassay system was significantly greater than that of the non-magnetic separation module. The magnetic separation module can obtain higher signal values in both low- and high-concentration samples, which lays a foundation for the detection of low-abundance clinical samples. 

In addition, the introduction of the magnetic separation module improved the detection repeatability of the electrochemiluminescence immunoassay system [Figure 1C,D]. We first repeated 10 tests on AFP quality control samples with high and low concentrations and found that the CVs obtained by the electrochemiluminescence system with magnetic separation were 0.5% and 1.3%, respectively, after 10 repeated measurements of low-concentration samples. Both percentages were significantly greater than those without magnetic separation (1.8% and 2.0%). To further demonstrate the improvement in the detection repeatability of the magnetic separation module, we carried out the same measurements with clinical samples and carried out 10 independent repeated measurements on the same clinical sample. The results are shown in Figure 1E,F for both high-concentration clinical samples and low-concentration clinical samples; the CV values (low-concentration samples: 0.5%, high-concentration samples: 1.5%) were significantly lower than those of non-magnetic separation electrochemiluminescence immunoassay systems (low-concentration samples: 0.8%, high-concentration samples: 1.7%).

### 3.2. A Compensation System for Current Output Photomultiplier Tubes to Improve Detection Sensitivity

Photomultiplier tubes are common detectors used in chemiluminescence signal detection techniques. Different chemiluminescence techniques differ in wavelength and signal strength due to the luminescent substances used, so photomultiplier tubes are selected for different signals. Typically, electrochemiluminescence immunoassay systems have a logarithmic amplifier to broaden the detection range of the circuit and ensure a comparable signal-to-noise ratio for data conversion over a large dynamic range, with the anode load resistance (RL) in the negative high-voltage system of the photomultiplier. However, this approach is limited by logarithmic curve compression: for inputs smaller than a certain value, the circuit outputs are negatively saturated. As a result, deviations in the interindividual consistency of the compression curve of the logarithmic amplifier circuit exist, which affects the results of small-signal measurements. We therefore optimized the design of a compensation system for current output photomultipliers to complement the variable signal with current to stabilize the input and ensure small signal detection capability.

The compensation circuit is a variable constant current source, and the constant current circuit is shown in Figure 2A. It consists of an adjustable voltage source and a high-value resistor. The adjustable voltage source included a reference voltage source and a potentiometer. The adjustable source is generated by dividing the voltage of the potentiometer against the reference supply. A constant current source has a bipolar compensation capability and provides a current in the same direction as the photomultiplier current when the photomultiplier dark current is less than the minimum input for the amplifier circuit design. When the photomultiplier dark current is greater than the minimum current of the amplification circuit, the constant current source provides a compensation current in the opposite direction to the photomultiplier current. The compensation current provided by the compensation circuit is superimposed on the photomultiplier quiescent current so that the quiescent current input to the logarithmic amplifier circuit is constant for the purpose of selecting a suitable operating range for the logarithmic amplifier.

In the electrochemiluminescence immunoassay system used in this study, the measured luminescence area of the electroluminescence measuring cell was approximately 4 × 5 cm, which was determined by the distribution of magnetic beads and the opposite electrode/working electrode region. In theory, the closer the luminescent body, the greater the light collection efficiency, and the larger the photosensitive area of the photomultiplier tube, the greater the efficiency. Owing to the constraints of device cost and volume, H10721 (HAMAMATSU PHOTONICS K.K., Shizuoka, Japan) was finally selected as the photomultiplier tube used in the system. In addition, this type of photomultiplier tube has a multibase photocathode in the near-infrared region and has high sensitivity. H10721 has four kinds of current output photomultiplier tubes, which are suitable for the detection of different wavelengths. In the electrochemiluminescence immunoassay system, the central wavelength of the tripyridine ruthenium luminescence used is 620 nm, so H10721-20 (HAMAMATSU PHOTONICS K.K., Shizuoka, Japan) and H10721-01 (HAMAMATSU PHOTONICS K.K., Shizuoka, Japan) are more suitable for this electrochemical luminescence system. At the same time, the sensitivity of H10721-20 is high, and the sensitivity curve is flat near the central wavelength. However, the dark current of H10721-20 is greater than that of H10721-01, and the range is relatively large [Figure 2B]. The electroluminescence is sensitive to low values, the dark current ratio is large and fluctuations affect the low SNR. Since the photomultiplier tube is a current output, the H10721-01 photomultiplier tube can be used directly in electrochemical luminescence immunoassays without the need for filters. In the tripyridine ruthenium–tripropylamine homogeneous solution test, the linear range of the system covered 10^5^, and saturation occurred when the concentration was greater than 1 μM. The maximum output of the 0.5 μM photomultiplier was 65.14 μA, and the output of the 0.01 nM photomultiplier was 1.3 nA, which was close to the dark current of the photomultiplier. The optimal interval is 0.1 nM~0.5 μM, and the integral signal of the luminous interval is approximately 1000 to 5,000,000 [Figure 2C].

Therefore, for the selection and testing of photomultiplier tubes, H10721-01 was selected as the photomultiplier device for the electrochemiluminescence immunoassay system. The design of the compensation circuit greatly improved the reception and detection of low detection signals, increasing the detection range of the electrochemiluminescence immunoassay system and providing it with the possibility to detect samples with low target substances.

### 3.3. Customized System Timing Module for High Test Throughput

After each component of the electrochemiluminescence immunoassay system was optimized, the properties of each component were maximized. Moreover, in order to achieve high-throughput results, the electrochemiluminescence immunoassay system requires optimization and precise control of the various steps in the analysis process to reduce various conflicts. The timing system module is a prerequisite for the orderly collaboration of the various parts of the analysis process, helps reduce conflicts between different actions, ensures the accuracy of the respective execution of parallel structures and is important for achieving high-throughput results.

Different scheduling algorithms are often used for different analytic systems (Table 1), and common scheduling algorithms include first-come, first-served algorithms [9,10], the shortest job algorithm [11,12,13], the shortest remaining time first algorithm [14,15,16] and the time-slice rotation algorithm [17]. The time-slice rotation algorithm is a system that allocates time slices to allocate resources to corresponding tasks in a fixed-length sequence. When the execution time of a task is complete, the system terminates the execution of that task and allocates the resources to the next task until all tasks have been executed. The time-slice rotation algorithm-based timing system therefore has good potential for application in multistep, multicomponent, multisystem synergistic electrochemiluminescence immunoassay systems.

On this basis, we used a time-slice rotation algorithm to customize the timing system for the electrochemiluminescence immunoassay system, effectively combining all parts of the system to facilitate the generation of high-throughput results. The process of electrochemiluminescence immunoassay analysis requires the synergy of multiple systems, such as potential generation, fluid control, permanent magnet control and analysis reagent switching. The accuracy of the timing and the rationality of the parameters and of each system are highly demanding. On the basis of the requirements of the electrochemiluminescence analysis task, we designed constant potential timing, fluid timing and permanent magnet timing in concert [Figure 3]. The design and customization of the timing of the system provides pipeline control of the various components of the electrochemiluminescence immunoassay system in accordance with the time rhythm, increasing the detection speed of the instrument to a maximum of 300 tests/hour, achieving a maximum unattended testing volume of 5.5 h and a maximum of 1575 samples, and ensuring a high degree of consistency between the luminescence moment and the electrode state, thus achieving a high-throughput result output. This enables high-throughput results to be output to meet the requirements of clinical applications.

### 3.4. Analytical Performance of the Electrochemiluminescence Immunoassay System

In this study, we validated the detection performance of the developed electroluminescence immunoassay system by detecting these three disease markers.

To evaluate the potential of an optimally designed high-throughput, high-sensitivity electrochemiluminescence immunoassay system for clinical application, we investigated the linearity, detection limits, signal-to-noise ratio, precision and accuracy of the system. First, we evaluated the linear relationship of the electrochemiluminescence immunoassay system. As shown in Figure 4A, as the concentration of the detected ACTH increased, the signal generated by the system also continuously increased. Even if the concentration exceeded 105 orders of magnitude, there was still an excellent linear relationship between the detected signal and the concentration (y = 265.9x + 10389, R^2^ = 0.9972), so this analysis system has excellent potential for detecting samples with different concentrations. We subsequently analyzed and evaluated the detection limits of the constructed electrochemiluminescence immunoassay system. We tested two detection limit samples (four repetitions for each sample) on 3 test days and statistically analyzed the test results. As shown in Figure 4B, ≥LoB (0.2 IU/mL) accounted for ≥87% of the total number of samples at the detection limit. Furthermore, five low-value samples with approximate detection limits (LoD ≤ 5 pg/mL) were tested, each sample was tested five times and the detection limits of the analysis system were evaluated. The results show that the number of test results below the blank limit (LoB ≤ 3 pg/mL) was less than or equal to 3, which meets the test requirements [Figure 4C]. Based on the detection limits of the above two different analytical systems, the constructed electrochemiluminescence immunoassay system has a good detection limit and has the ability to detect low-abundance clinical samples.

The signal-to-noise ratio is another important index for evaluating the detection capability of an electrochemiluminescence immunoassay system and reflects the strength of the received useful signal and the received interference signal. In this study, the signal-to-noise ratios of the constructed electrochemiluminescence immunoassay system for the detection of clinical samples containing AFP and hs-cTnT-STAT were compared with the SNR of the international leading international electrochemiluminescence immunoassay instrument of Roche. As shown in Figure 5, the constructed system has a better signal-to-noise ratio when detecting both AFP and hs-cTnT-STAT, and there is a significant difference even when detecting low-concentration samples. Therefore, when detecting AFP and hs-cTnT-STAT, the constructed electrochemiluminescence immunoassay system has an excellent signal-to-noise ratio, good sensitivity and excellent detection performance.

To evaluate whether the constructed electrochemiluminescence immunoassay system can be used for the detection of clinical samples, we further analyzed its detection precision. First, the samples containing high-concentration and low-concentration hs-cTnT STAT substances were tested independently 40 times via two sets of electrochemiluminescence immunoassay systems [Figure 6A]. The statistical results revealed that when standard substance H was detected, the CV values of the two systems were 4.02% and 4.70%, respectively. The CVs of standard substance L were 3.46% and 3.62%, respectively, and both were <5%, meeting the requirements. Furthermore, we examined different clinical samples from 20 consecutive days to analyze their precision over time. As shown in Figure 6B, both sets of analysis systems achieved excellent detection precision for different samples during the 20-day detection period. When clinical sample H was tested, its CV was 4.67% and 3.13%, and when clinical sample L was tested, its CV was 4.92% and 2.86%, respectively. Therefore, the excellent detection precision of the constructed electrochemiluminescence immunoassay system was determined by evaluating the reproducibility of the measurements.

Finally, the detection accuracy of the constructed electrochemiluminescence immunoassay systems was evaluated. The detection results for AFP [Figure 7A] and hs-cTnT-STAT [Figure 7B] which were obtained by the system were compared with those of the Roche electrochemiluminescence immunoassay instrument. First of all, when the analysis system detected AFP and hs-cTnT-STAT, its detection results were in good agreement with those of the international Roche system. The correlations between the detection results of the two systems were 0.9976 and 0.999; thus, there was no significant difference between the detection results of the two systems. Therefore, the constructed electrochemiluminescence immunoassay system has excellent detection accuracy and can reach the leading international level. The relative deviation, absolute deviation, relative deviation ranking and absolute deviation ranking of the AFP detected by the system were analyzed. As shown in Figure 7C–F, both the relative deviation and the absolute deviation reflect the detection accuracy of the analysis system.

In summary, the high-throughput and high-sensitivity electrochemiluminescence immunoassay system shows superior performance in terms of the following basic properties: online relationship, detection limit, signal-to-noise ratio, precision and accuracy. Therefore, the system is expected to become an alternative method for clinical analysis, achieving high throughput and high sensitivity, and provide a reference for clinical disease diagnosis.

## 4. Conclusions

In this work, the magnetic separation module, photomultiplier and timing system module of the electrochemiluminescence immunoassay system were optimized to meet the requirements of clinical testing for detection accuracy, detection sensitivity and result throughput. First, to reduce the influence of nontarget substances in the sample, a magnetic separation module for the electrochemiluminescence immunoassay was constructed using magnetic beads and permanent magnets. The module was used to purify and enrich the target substances, improving the detection accuracy of the system and improving the detection repeatability of the system. This was followed by the design of a compensation circuit for the current-type photomultiplier to address the shortcomings of the conventional single photomultiplier in detecting minute signals, enabling the detection of low sample and low target-substance contents and improving the sensitivity of the electrochemiluminescence immunoassay system. Second, on the basis of the time-slice rotation algorithm, the timing system was customized according to the timing requirements of the electrochemiluminescence immunoassay system for high-throughput detection of target substances, achieving a maximum testing speed of 300 tests/hour. Finally, the optimized electrochemiluminescence immunoassay system was analyzed for linearity, detection limit, signal-to-noise ratio, precision and accuracy, and the results showed superior performance. In summary, by optimizing the throughput output, detection accuracy and detection sensitivity, a high-throughput and high-sensitivity electrochemiluminescence immunoassay system was successfully constructed and has shown great potential for clinical application.

## Figures and Tables

**Figure 1 bioengineering-11-00885-f001:**
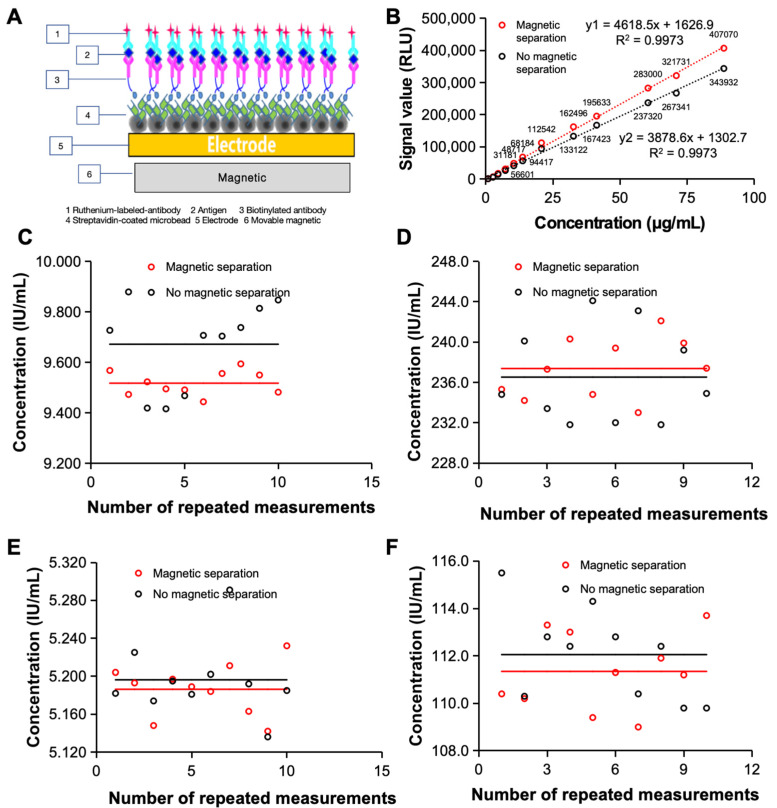
A magnetic separation module was used to increase the detection accuracy. (**A**) Schematic diagram of magnetic separation. (**B**) Fitting curve between the concentration of the FDP and the output signal of the electrochemiluminescence immunoassay system. The repeatability of the standard was tested independently 10 times (**C**) with low concentrations and (**D**) high concentrations of the target analyte in the sample. The repeatability of the clinical sample was tested independently 10 times (**E**) at low concentrations and (**F**) at high concentrations for the targets in the sample.

**Figure 2 bioengineering-11-00885-f002:**
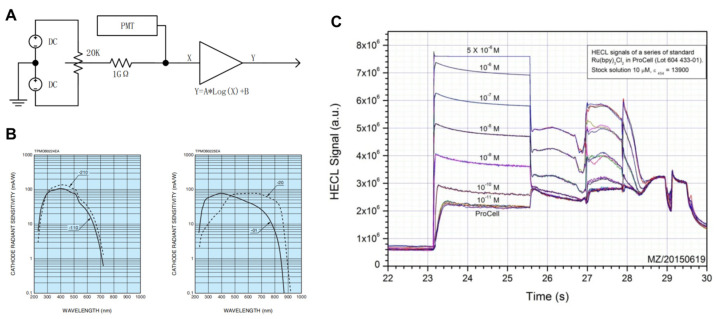
A photomultiplier tube system was used to increase the detection sensitivity. (**A**) Schematic block diagram of the compensation circuit. (**B**) Wavelength sensitivity curve of H10721. (**C**) Signal output curve of the electrochemiluminescence immunoassay system.

**Figure 3 bioengineering-11-00885-f003:**
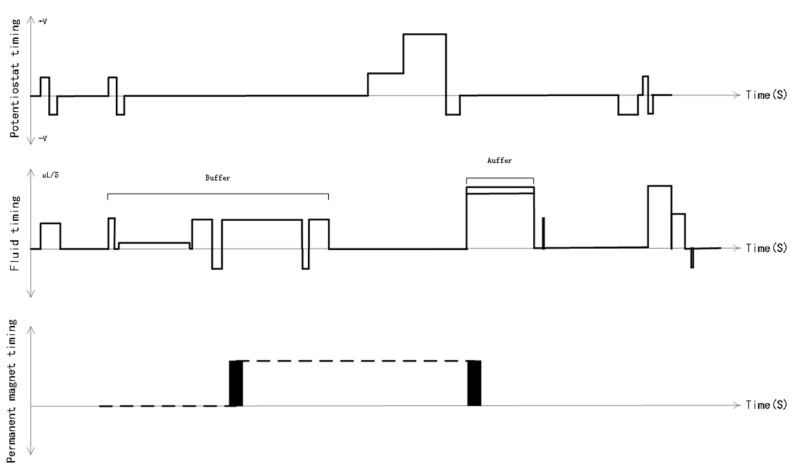
Time series of the electrochemiluminescence immunoassay system used to improve test throughput.

**Figure 4 bioengineering-11-00885-f004:**
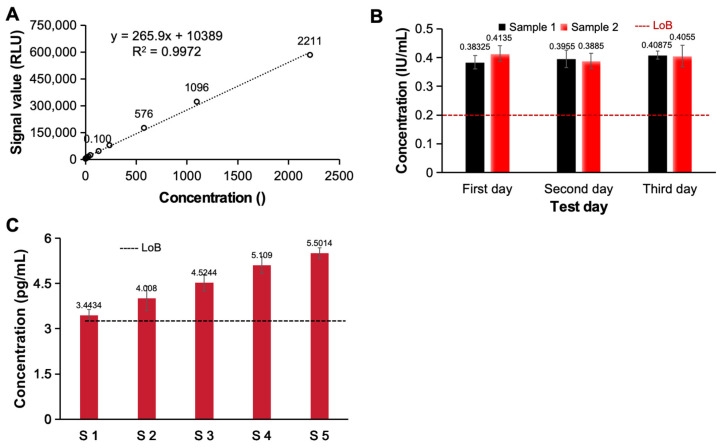
The detection performance of this electrochemiluminescence immunoassay system. (**A**) The linearity of this system for ACTH detection. The detection limitations of this system for AFP (**B**) and hs-cTnT-STAT (**C**).

**Figure 5 bioengineering-11-00885-f005:**
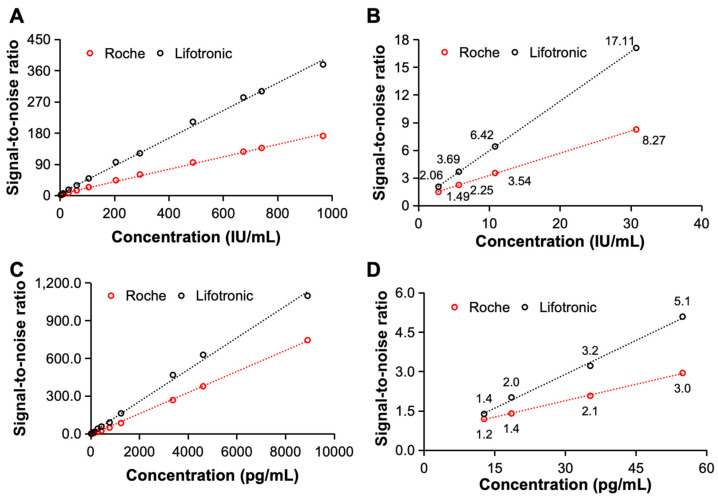
The signal-to-noise ratios of the proposed electrochemiluminescence immunoassay system. (**A**,**B**) The signal-to-noise ratio of the proposed system and that of Roche for AFP detection. (**C**,**D**) The signal-to-noise ratio of the proposed system and that of Roche for detecting hs-cTnT-STAT.

**Figure 6 bioengineering-11-00885-f006:**
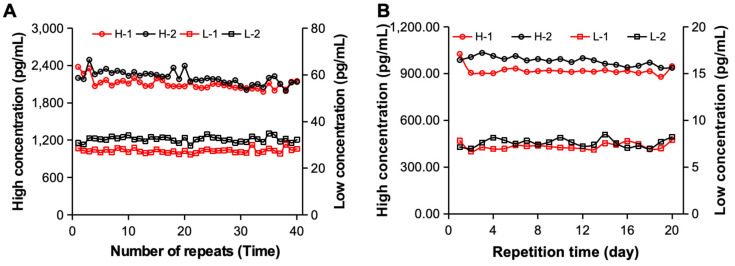
The detection precision of the proposed electrochemiluminescence immunoassay system. (**A**) Results of 40 independent repeated measurements of samples containing different concentrations of hs-cTnT STAT by two independent electrochemiluminescence immunoassay systems. (**B**) Results of 20 days of continuous monitoring of samples containing different concentrations of hs-cTnT STAT by two independent electrochemiluminescence immunoassay systems.

**Figure 7 bioengineering-11-00885-f007:**
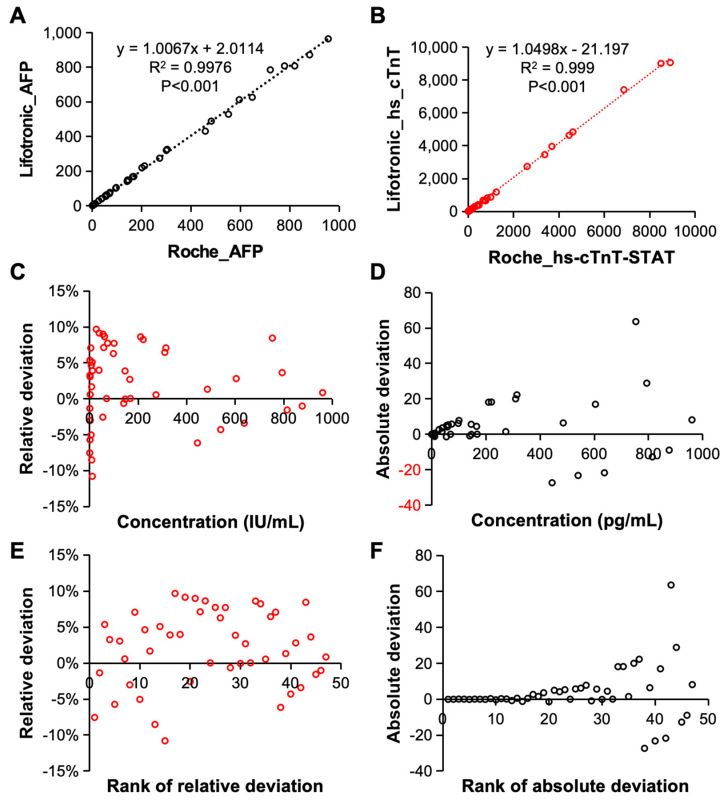
The detection accuracy of this electrochemiluminescence immunoassay system. When AFP (**A**) and hs-cTnT-STAT (**B**) were detected, the correlation between the test results of the system and the results of the Roche system was evaluated. The relative deviation (**C**), absolute deviation (**D**), relative deviation ranking (**E**) and absolute deviation ranking (**F**) of the AFP detected by the system.

**Table 1 bioengineering-11-00885-t001:** Comparison of the advantages and disadvantages of each algorithm.

Algorithms	Advantages	Restrictions	Reference
First come, first served	Simplest and easy to understand Easy to run in the program Fair for all tasks	More time-consuming Low resource utilization	[9,10]
Shortest job priority	Improved average turnaround times Reduced waiting time for tasks Increases system throughput	Need to anticipate run times Each task needs to run independently and unrelated Not good for long tasks	[11,12,13]
Shortest remaining time priority	Can provide good service for new short tasks	Longer tasks are more likely to be interrupted	[14,15,16]
Time-slice rotation algorithm	Combines long and short tasks Effective resource utilization and efficiency	Context switching is complex Time-slice size selection has a significant performance impact	[17]

## Data Availability

All data for the study can be obtained by contacting G.W. (wanggx@cqu.edu.cn).

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
