# Peer review of "A Novel High-Throughput and Sensitive Electrochemiluminescence Immunoassay System"

_bioengineering, 2024, doi:10.3390/bioengineering11090885_

Round 1

Reviewer 1 Report

Comments and Suggestions for Authors

The manuscript presents an interesting analytical development, but quality of data presentation and interpretation is not acceptable for publication. In my opinion, the manuscript should be basically re-written and re-submitted for «de novo» estimation.

- First of all, the subject of investigation completely accords to thematic field of the Biosensors journal, and re-addressing to it seems reasonable.

- The work should be structured as clear application of the proposed approach to ACTH and/or hs-cTNT detection. It should collect actual needs for assaying these compounds, reached analytical parameters including data about real samples testing, consideration of competing developments and grounding the reached advantages.

- The Abstract should be focused on specific features of the proposed system, and the reached quantitative parameters of the assays. Non-grounded estimation and discussion of successes of Roche and other companies should be excluded.

- The Introduction should clearly specify the place of the proposed assay among the existing electrochemiluminescent immunoassays, the authors' ideas that were implemented in the first time.

- The text should be checked to exclude poorly grounded overall estimations such as table 1 without addressing to data of comparative investigations. Self-complimentary statements could be accepted only with their confirmations by clear quantitative comparisons with alternate techniques.

Author Response

Response to Reviewer #1:

Comments #1:

The manuscript presents an interesting analytical development, but quality of data presentation and interpretation is not acceptable for publication. In my opinion, the manuscript should be basically re-written and re-submitted for «denovo» estimation.

- First of all, the subject of investigation completely accords to thematic field of the Biosensors journal, and re-addressing to it seems reasonable.

Response:

Thanks for your opinion, this study focused on the development and optimization of the electrochemiluminescence immunoassay testing instrument. Specifically, the text is mainly about optimizing the detection capacity with high throughput and high accuracy performance by designing magnetic separation modules to increase the detection accuracy and introducing PMT to increase the detection sensitivity and finally optimizing the system timing regulation capability to realize maximum test output. Thus, in our opinion, this study in line with the requirement and position of the journal of “bioengineering”.

Comments #2:

- The work should be structured as clear application of the proposed approach to ACTH and/or hs-cTNT-STAT detection. It should collect actual needs for assaying these compounds, reached analytical parameters including data about real samples testing, consideration of competing developments and grounding the reached advantages.

Response:

Thanks for your suggestions. We have added relative text in the manuscript in Page.

In this study, we wanted to optimize the performance of the electrochemiluminescence immunoassay testing instrument to meet the requirements of the clinic analysis. Thus, to further analysis the capability of the instrument, we used the optimized equipment to analyze ACTH or hs-cTNT-STAT and compare the main analysis performance with the same instrument in Roach company, which is the gold standard in electrochemiluminescence immunoassay testing instrument. Therefore, we did not emphasize in this paper the actual need to analyze ACTH and hs-cTNT-STAT and the data related to actual sample testing.

We choose two analysis sample with two reasons, on the one hand, we can test whether the detection capability of the optimized equipment meets the requirements. On the other hand, we can confirm that the equipment can meet the requirements of clinical detection of various substances and has good universality by detecting various substances.

In order to determine the competitive development and achieved advantages of the electrochemiluminescence immunoassay equipment proposed in this study, we mainly carried out comparative experiments and compared it with the gold standard (related equipment in Roche company) to achieve the analysis of the advantages achieved in this study.

Moreover, in this article, we do not focus on the detection of ACTH and/or hs-cTNT-STAT detection methods. This is because our proposed electrochemiluminescence immunoassay instrument has the universality to meet the clinical needs, and it can be detected by simply changing the corresponding matching detection reagents.

Comments #3:

- The Abstract should be focused on specific features of the proposed system, and the reached quantitative parameters of the assays. Non-grounded estimation and discussion of successes of Roche and other companies should be excluded.

Response:

Thanks for your advice. Indeed, in the abstract we should focus on describing the characteristics of our own system and the measurement data achieved. We have revised the corresponding content and re-wrote the abstract, focusing on the optimization and achievements of our own system. Page 1 and line 19-35.

Comments #4:

- The Introduction should clearly specify the place of the proposed assay among the existing electrochemiluminescent immunoassays, the authors' ideas that were implemented in the first time.

Response:

Thanks for your suggestion, we have made corresponding additions in the manuscript. Page 3 and line 83-87.

By optimizing the system composition and system design, it provides the detection throughput, sensitivity and accuracy of the whole system. This study is to accelerate the clinical application of electrochemiluminescence immunoassay technology, which is helpful for the rapid and sensitive detection of clinical markers.

Comments #5:

- The text should be checked to exclude poorly grounded overall estimations such as table 1 without addressing to data of comparative investigations. Self-complimentary statements could be accepted only with their confirmations by clear quantitative comparisons with alternate techniques.

Response:

Thank you for your comments and suggestions. In this paper, we summarize the relevant algorithms and their advantages and limitations in table 1. Due to our mistakes, we only cited relevant references in the description part of the manuscript, but did not cite relevant literature in the table, which caused misunderstanding. We have added relevant references to the table to further strengthen the scientific and accuracy of the study. Page 10.

Reviewer 2 Report

Comments and Suggestions for Authors

The presented work is on an electrochemiluminescent immunoassay device. The overall study is interesting. A few corrections have to be made before considering it for publication. 

Please address the following comments:

1. In this study, a magnetic separation module was used to improve the detection accuracy. The authors must mention how the magnetic beads were functionalized with the biomolecules. Details on the methods and materials have to be given in this section. 

2. Figure 1A represents the Schematic diagram of magnetic separation. However, the components in the figure are not marked. It has to be marked.

3. Methods should be mentioned in the “materials and method” and not under the result. 

4. Give more description of the bio-samples used. Ex: instead of just writing AFP, give more details of it.

5. Units are missing in the axes of graphs. Please correct it.

Comments on the Quality of English Language

Language is fine

Author Response

Response to Reviewer #2:

Comments and Suggestions for Authors

The presented work is on an electrochemiluminescent immunoassay device. The overall study is interesting. A few corrections have to be made before considering it for publication. 

Please address the following comments:

Comments #1:

  1. In this study, a magnetic separation module was used to improve the detection accuracy. The authors must mention how the magnetic beads were functionalized with the biomolecules. Details on the methods and materials have to be given in this section. 

Response:

Thank you for your advice. In fact, we purchased streptavidin labeled magnetic beads directly from established companies. Detailed information: Dynabeads M-280 Streptavidin were purchased from invitrogen (35131D). Page 17, line 90-94.

Comments #2:

  1. Figure 1A represents the Schematic diagram of magnetic separation. However, the components in the figure are not marked. It has to be marked.

Response:

Thank you for your advice. This was our mistake and we have added the relevant information to the Figure 1A. Page 6.

Comments #3:

  1. Methods should be mentioned in the “materials and method” and not under the result. 

Response:

Thank you for your advice. We switched the method to the section of Materials and methods, and revised the manuscript accordingly. Page: 1, line 26-33.

Comments #4:

  1. Give more description of the bio-samples used. Ex: instead of just writing AFP, give more details of it.

Response:

Thank you for your advice. We have added detailed information including full name of AFP, source of AFP already used in this study, article number and other details. And the similar problems have been examined and revised.

Alpha Feto Protein is the full name of the AFP , and the AFP used in this study is from the WHO NIBSC, article numbers SRM@2921 and 1st 72/225.

Comments #5:

  1. Units are missing in the axes of graphs. Please correct it.

Response:

Thank you for your questions. This is our problem, we added the units of the axis of the figure. And the similar problems have been examined and revised.

Round 2

Reviewer 1 Report

Comments and Suggestions for Authors

The authors have revised their manuscript and made significant improvements.

However, in my opinion, the work, while presenting the same array of experimental data in the same form, retains the main drawback noted during the first review: "The work should be structured as a clear application of the proposed approach to ACTH and/or hs-cTNT detection. It should collect actual needs for assaying these compounds, reached analytical parameters including data about real samples testing, consideration of competing developments and grounding the reached advantages."

In the proposed inserts, the authors declare the importance of these issues and their interest in them, but the presented material still does not describe a complete analytical development with an assessment of its capabilities and competitive potential, but a certain set of data related to the use of a modified electrochemiluminescent immunoassay format. Additional references for comparison with alternative methods are considered only at the level of qualitative comments, without providing a comparison in sensitivity, expressivity, other significant parameters.

Unfortunately, I still cannot recommend the submitted manuscript for publication.

Author Response

Dear Editor,

We gratefully appreciate for reviewers and editors spending precious time in making constructive remarks to our manuscript entitled “Novel high-throughput and sensitive electrochemiluminescent immunoassay system”, [Research Article, manuscript ID: bioengineering-3126426-R1]. We have carefully considered the comments for editor and reviewers, which help us to improve the manuscript substantially and guide significance to our studies. All comments and suggestions were accurately incorporated and considered.

The changes in the revised manuscript are highlighted in "red". We also provide a point-to-point response below, referencing the changes in the text, for your convenience to check (see our response in "blue" below). After the revision, we believe that the quality of our manuscript was significantly improved and could meet the requirements of your journal.

Guixue Wang, Ph.D,FBSE

Professor & Directors

MOE Key Lab for Biorheologic

Sci & Tech, National Local Joint

Eng Lab for Vascular Implants

College of Bioengineering

Chongqing University

Chongqing 400044, China

Email: wanggx@cqu.edu.cn

Response to Reviewer #1:

Comments #1:

Comments and Suggestions for Authors

The authors have revised their manuscript and made significant improvements. 

However, in my opinion, the work, while presenting the same array of experimental data in the same form, retains the main drawback noted during the first review: "The work should be structured as a clear application of the proposed approach to ACTH and/or hs-cTnT detection. It should collect actual needs for assaying these compounds, reached analytical parameters including data about real samples testing, consideration of competing developments and grounding the reached advantages." 

In the proposed inserts, the authors declare the importance of these issues and their interest in them, but the presented material still does not describe a complete analytical development with an assessment of its capabilities and competitive potential, but a certain set of data related to the use of a modified electrochemiluminescence immunoassay format. Additional references for comparison with alternative methods are considered only at the level of qualitative comments, without providing a comparison in sensitivity, expressivity, other significant parameters. 

Unfortunately, I still cannot recommend the submitted manuscript for publication.

Response:

Thanks for your comments and suggestions, we have further added the content you mentioned in detail in the manuscript to enrich our content.

Adrenocorticotropic hormone (ACTH) is a 39-amino acid peptide hormone (4.5 kDa) released from the anterior pituitary gland [1]. ACTH regulates corticosteroid hormone production, which has important functions in a myriad of critical physiological functions [2,3,4]. For early and accurate assessment of altered ACTH secretion, advances in its detection are required; however, there are challenges associated with the diagnosis for altered ACTH level. The fluctuation of ACTH in serum (<4.1 to 51.4 pg/mL) makes diagnosis even more problematic [5]. To address these challenges in diagnosis, a rapid, sensitive, and selective detection method is needed.

Acute myocardial infarction (AMI) is an acute coronary syndrome in which blocked coronary artery causes insufficient blood supply to the heart muscle and impairs heart function, which is the leading cause of cardiovascular death worldwide.[6] Rapid and accurate early diagnosis of AMI is the key to reduce mortality and improve prognosis of patients.[7] Cardiac troponin (cTn) has become a traditional recognized indicator for the diagnosis of AMI,[8] since cTn lacks sensitivity in the early stages of myocardial injury. The sensitivity of hypersensitive cardiac troponin (hs-cTnT) is higher than that of traditional troponin. Therefore, it is necessary to develop a new technique to realize the high sensitive detection of hs-cTnT and strengthen the early diagnosis of AMI.

Therefore, in this study, we validated the detection performance of the developed electrochemiluminescence immunoassay system by detecting these two disease markers.

In order to explain the detection performance of the electrochemiluminescence immunoassay system constructed by us. We use a comparative method to determine its detection performance. Even though I extracted the data generated by my own system during the detection, in order to show the advantages of the data, we finally compared it with the standard instrument method of Roche company, which we believe can better show the accuracy and importance of the method we used.

In the last part, we use ACTH and/or hs-cTNT as detection targets. The opportunity detection capability of the electrochemiluminescence immunoassay system in linearity, detection limits, signal-to-noise ratio, precision and accuracy was analyzed. The obtained test results all meet the current clinical needs and have a good consistency with the data of Roche's testing instruments currently used for clinical testing, and some test data are superior to them. Thus, we believe that the electrochemiluminescence immunoassay system developed by us has excellent detection performance, after all, good clinical application potential and market competitive advantages.

Reference:

  1. White, A.A. ACTH: Cellular peptide hormone synthesis and secretory pathways. Results Probl. Cell Differ.2010, 50, 63–84.
  2. Duda, T.; Pertzev, A.; Rameshwar, K.S. Ca(2+) modulation of ANF-RGC: New signalling paradigminterlocked with blood pressure regulation. Biochemistry 2012, 51, 9394–9405.
  3. Varadhan, L.; Aror, A.; Walker, A.B.; Varughese, G.I. Cushing’s disease: Establishing the diagnosis andmanagement approach. J. Assoc. Physicians India 2013, 61, 278–280.
  4. Hale, A.C.; Besser, G.M.; Rees, L.H. Characterization of pro-opiomelanocortin-derived peptides in pituitaryand ectopic adrenocorticotrophin-secreting tumours. J. Endocrinol. 1986, 108, 49–56.
  5. 5. Rees, L.H.; Cook, D.M.; Kendall, J.W.; Allen, C.F.; Kramer, R.M.; Ratcliffe, J.G.; Knight, R.A. A radioimmunoassayfor rat plasma ACTH. Endocrinology 1971, 89, 251–254.
  6. Goodacre S, Cross E, Arnold J, et al. The health care burden of acute

chest pain. Heart (British Cardiac Society) 2005;91:229–30.

  1. Fan J, Ma J, Xia N, et al. Clinical Value of Combined Detection of CKMB, MYO, cTnI and Plasma NT-proBNP in Diagnosis of Acute

Myocardial Infarction. Clin Laboratory 2017;63:427–33.

  1. Ray P, Charpentier S, Chenevier-Gobeaux C, et al. Combined copeptin

and troponin to rule out myocardial infarction in patients with chest pain

and a history of coronary artery disease. Am J Emergency Med

2012;30:440–8.

Reviewer 2 Report

Comments and Suggestions for Authors

All questions are answered

Comments on the Quality of English Language

Minor corrections are required

Author Response

Response to Reviewer #2:

Comments #1:

Comments on the Quality of English Language

Minor corrections are required

Response:

Thanks for your opinion, we have revised the overall language of the article. We believe that the revised language can meet the high requirements of the magazine.

Round 3

Reviewer 1 Report

Comments and Suggestions for Authors

The actual version of the manuscript contains key data characterizing analytical parameters of the proposed biosensor. Although I do not consider the form of their presentation to be completely clear and convenient, the article can nevertheless be approved for publication.